# Enzyme Treatment Alters the Anti-Inflammatory Activity of the Water Extract of Wheat Germ In Vitro and In Vivo

**DOI:** 10.3390/nu11102490

**Published:** 2019-10-16

**Authors:** Youngju Song, Hee-Young Jeong, Jae-Kang Lee, Yong-Seok Choi, Dae-Ok Kim, Davin Jang, Cheon-Seok Park, Sungho Maeng, Hee Kang

**Affiliations:** 1Department of Biomedical Science and Technology, Graduate School of Kyung Hee University, Seoul 02447, Korea; songjy@khu.ac.kr; 2Graduate School of East-West Medical Science, Kyung Hee University, Yongin 17104, Korea; jhy9592@naver.com (H.-Y.J.); jethrot@khu.ac.kr (S.M.); 3Sajo DongA One, Dangjin 31703, Korea; macmen@sajo.co.kr (J.-K.L.); knownet@kodoco.com (Y.-S.C.); 4Department of Food Science and Biotechnology, College of Life Sciences, Kyung Hee University, Yongin 17104, Korea; dokim05@khu.ac.kr (D.-O.K.); davin1031@khu.ac.kr (D.J.); cspark@khu.ac.kr (C.-S.P.); 5Humanitas College, Kyung Hee University, Yongin 17104, Korea

**Keywords:** wheat germ, enzyme, benzoquinone, inflammation, macrophages

## Abstract

Wheat germ is rich in quinones that exist as glycosides. In this study, we used Celluclast 1.5L to release the hydroxyquinones, which turn into benzoquinone, and prepared the water extract from enzyme-treated wheat germ (EWG). We investigated whether enzyme treatment altered the anti-inflammatory activity compared to the water extract of untreated wheat germ (UWG). UWG inhibited the production of inducible nitric oxide synthase (iNOS) and interleukin (IL)-12 and induced the production of IL-10 and heme oxygenase (HO)-1 in lipopolysaccharide (LPS)-stimulated macrophages. Enzyme treatment resulted in greater inhibition of iNOS and IL-10 and induction of HO-1 compared to UWG, possibly involving the modulation of nuclear factor (NF)-κB, activator protein 1 (AP-1) and nuclear factor erythroid 2-related factor (Nrf2). Mice fed UWG or EWG had decreased serum tumor necrosis factor (TNF)-α and increased serum IL-10 levels after intraperitoneal injection of LPS, with UWG being more effective for IL-10 and EWG more effective for TNF-α. Hepatic HO-1 gene was only expressed in mice fed EWG. We provide evidence that enzyme treatment is a useful biotechnology tool for extracting active compounds from wheat germ.

## 1. Introduction

Wheat is one of the major cereals consumed worldwide. Wheat consists of the endosperm (80–85%), germ (2.5–3%), and bran (10–14%) [1]. Compared to the endosperm and bran fractions, wheat germ contains higher proportions of sulfur compounds, dietary fibers (lignins, oligosaccharides, and phytic acids), B vitamins, vitamin E, phytosterols (mainly sitosterol and campesterol), flavonoids, total choline, vitamin E, flavonoids, and choline [1]. Wheat germ is specifically rich in linoleic acid, which is susceptible to oxidative reactions and thus affects the long-term storage of wheat flour quality [2]. Also, wheat germ has a high protein content and is considered as a source of good vegetable proteins [2]. Despite its nutritional value, the major use of wheat germ generated during milling is limited to animal feed. It is estimated that approximately 25,000,000 tons of wheat germ are produced annually [3]. Therefore, it is important to determine the heath-enhancing potential of wheat germ after applying various processing technologies.

Wheat germ is reported to be one of the natural products rich in methoxyhydroquinones and dimethoxyhydroquinones, which exist as beta-glucosides [4]. When these hydroquinones are released via the breakage of β-glucosidic bonds, they are oxidized to 2-methoxy benzoquinone (MBQ) and 2,6-dimethoxy-benzoquinone (DMBQ). Wheat germ possesses lactic acid bacteria, which display β-glucosidase activity at 30 ℃ and release hydroquinones under appropriate conditions [3,5]. Fermentation can increase the release of hydroquinones in wheat germ [5,6]. Several studies have shown that the water extract of wheat germ fermented with yeast exhibits anti-cancer effects [4,7,8,9], and the extract is commercially available as a nutritional supplement for cancer patients. Hidvegi, who first patented the water extract of fermented wheat germ, established MBQ and DMBQ as quality markers for its manufacture. Some studies have suggested that MBQ and DMBQ are the active compounds in fermented wheat germ extract, but other studies have shown that these benzoquinones do not affect its biological activities [6,10,11]. 

Inflammation is required for the host to clear noxious stimuli and repair the subsequent tissue damage. Macrophages are tissue-resident immune cells that detect the presence of harmful substances such as lipopolysaccharides (LPSs). Macrophages initiate this inflammatory response by producing inducible NO synthase (iNOS) and cyclooxygenase (COX)2 and pro-inflammatory cytokines such as tumor necrosis factor (TNF)-α, interleukin (IL)-6, and IL-12 through the nuclear factor-κB (NF-κB) and mitogen-activated protein kinase (MAPK) pathways [12,13]. In addition to their capacity to promote inflammation, macrophages provide a feedback response by producing their own anti-inflammatory proteins such as IL-10 and heme oxygenase (HO)-1 [14]. However, uncontrolled and inappropriate macrophage response may hamper the healing process of the host, leading to chronic inflammation.

Cereal-related products can be applied for the prevention and slow progression of chronic inflammatory diseases such as atherosclerosis and obesity. Cereal brans or some fermented cereal products have been shown to exert anti-inflammatory activity [15,16,17,18,19]. Previously, we obtained a water-soluble extract from wheat germ by applying citric acid hydrolysis at 60 ℃. This extract contained a small amount of DMBQ (0.065 mg/g), which was far below that present in standardized fermented wheat germ extract (0.4 mg/g) [20]. Although citric acid hydrolysis is economical, the application of a relatively high temperature proved to be suboptimal for generating DMBQ. Regardless of the DMBQ level, untreated wheat germ extract itself showed anti-inflammatory activity in vitro and citric acid treatment improved this activity [20]. However, if the amount of quinones increases, it is not clear whether the anti-inflammatory function of water-soluble wheat germ extract also increases. 

Celluclast 1.5 L is a commercially available cellulolytic enzyme preparation from *Trichoderma reesei* that has endo-1,4-β-D-glucanase, exo-1,4-β-D-glucanase, β-glucosidase, and β-xylosidase activities [21]. Therefore, it is hypothesized that treating wheat germ with Celluclast 1.5L will generate various aglycones including quinones released from small glycosides, resulting from degradation of cell wall components such as arabinoxylan. In this study, we prepared a water extract from Celluclast 1.5L-treated wheat germ (EWG) and investigated whether enzyme treatment enhanced the anti-inflammatory activity of untreated wheat germ extract (UWG) in vitro and in vivo. We also tried to determine whether DMBQ is an important ingredient for anti-inflammatory function of water-soluble wheat germ extract.

## 2. Materials and Methods 

### 2.1. Preparation of UWG and EWG

Wheat germ was provided by Sajo DongA One (Dangjin, South Korea). The germ was separated during the milling of *Triticum aestivum* (Australia Standard White Wheat). Unground wheat germ (50 g) was incubated in 250 mL of water with or without 0.5% Celluclast 1.5L (Novozymes, Bagsvaerd, Denmark) for 24 h at 30 ℃ in a water bath. After centrifugation, the supernatant was freeze-dried using a vacuum freeze-dryer (Eyela, Tokyo, Japan). 

### 2.2. HPLC Analysis of DMBQ

One gram of UWG or EWG was dissolved in 100 mL of deionized water then extracted three times by shaking for 1 min with 300 mL of chloroform. The chloroform layers were obtained and evaporated to dryness using a vacuum evaporator at 35 °C. The dried materials were redissolved in 10 mL of chloroform and filtered through a 0.45 μm polyvinylidene fluoride filter. The standard used for the analysis of DMBQ was purchased from Sigma (St. Louis, MO, USA). DMBQ was analyzed using an HPLC system (Alliance; Waters, Milford, MA, USA) equipped with a photodiode array detector (Waters) operating at 290 nm and a ProntoSIL 120-5-C_18_ ACE-EPS column (250 × 4.6 mm, 5μm). The injection volume was 5 μL and the flow rate was 0.8 mL/min. Eluent A consisted of 0.1% (v/v) formic acid in deionized water and eluent B was acetonitrile. The solvent compositions for the binary mobile phases were as follows, with each segment lasting 5 minutes: linear gradient 0–25% B, hold at 25% B, linear gradient 25–35% B, hold at 35% B, linear gradient 35–85% B, hold at 85% B, linear gradient 85–0% B, and hold at 0% B. 

### 2.3. Animals

Male Balb/c mice aged 7 weeks were purchased from Koatech (Pyungtek, South Korea) and underwent 1 week of adjustment prior to experiments. The animal protocol (KHUASP(GC)-19-005) was approved by the Institutional Animal Care and Use Committee of Kyung Hee University, and mice were cared for according to the specifications of the US National Research Council for the Care and Use of Laboratory Animals (1996).

### 2.4. Macrophage Isolation

Mice were injected intraperitoneally with 2 mL of 3.5% sterile thioglycollate (BD, Sparks, MD, USA). Four days later, the mice were sacrificed via CO_2_ inhalation and peritoneal exudate cells were harvested by injecting 7 mL of cold DMEM (HyClone, Logan, UT, USA) plus 1% fetal bovine serum (FBS; HyClone) and 1% penicillin-streptomycin. After centrifugation, the cells were resuspended in DMEM plus 10% FBS and 1% penicillin-streptomycin and counted using a Countess II Automated Cell Counter (Thermo Scientific, Bothell, WA, USA). The cells were then plated and incubated overnight at 37 ℃, and the non-adherent cells were removed. 

### 2.5. Cell Viability Assay

Cell viability was tested using an MTT assay. Cells in 96-well plates were incubated for 24 h with increasing concentrations of UWG, EWG and DMBQ and then the culture medium was removed. MTT (final concentration 0.5 mg/mL) (Sigma) was added to each well for 1 h, and then the media was removed. DMSO was added and incubated for 15 min to solubilize the MTT. Optical density was measured at 570 nm with an iMark microplate reader (Bio-Rad, Hercules, CA, USA). Cell viability was expressed as a percentage of control cells. 

### 2.6. Western Blotting

To detect iNOS, COX2, and HO-1, cells were stimulated with or without 100 ng/mL LPS (Sigma) and simultaneously incubated with UWG, EWG, or DMBQ for 24 h. For IκBα and phosphate MAPK, cells were pretreated with UWG, EWG, or DMBQ for 1 h and then stimulated with LPS for 15 min. For nuclear Nrf2, cells were incubated with UWG or EWG for the indicated time period. Whole cell lysates were prepared by resuspending the cells in RIPA buffer (50 mM Tris-HCl, pH 7.5; 150 mM NaCl; 1 mM EDTA; 20mM NaF; 0.5% NP-40; 1% Triton X-100) plus a phosphatase inhibitor cocktail (Sigma) and a protease inhibitor cocktail (Quartett, Berlin, Germany). Nuclear protein lysates were prepared using the Nuclear Extraction Kit (Active Motif, Carlsbad, CA, USA) according to the manufacturer’s protocol. Protein concentration was determined using the Bradford assay. Protein lysates were separated on an 8% or 10% sodium dodecyl sulfate-polyacrylamide gel and were then transferred to a polyvinylidene fluoride membrane. The membranes were blocked with 5% skim milk in Tris-buffered saline with 0.1% Tween 20 (TBST) for 1 h then incubated overnight at 4℃ with primary antibodies against iNOS, COX2 (Cayman Chemical, Ann Arbor, MI, USA), HO-1, IκBα, glyceraldehyde 3-phosphate dehydrogenase (GAPDH) (Santa Cruz Biotechnology, Santa Cruz, CA, USA), phospho-JNK, JNK, phospho-p38, p38, phospho-ERK1/2, ERK1/2, Nrf2, or lamin B1 (Cell Signaling Technology, CA, USA) diluted 1:1000 in 5% skim milk in TBST. The blots were washed with TBST and incubated for 1 h with anti-rabbit horseradish peroxidase-conjugated antibody (diluted 1:5000 in 5% skim milk in TBST). The protein bands were detected with EzWestLumi plus (ATTO, Tokyo, Japan) and analyzed using an EZ-Capture MG (ATTO).

### 2.7. Cytokine Analysis

The levels of TNF-α, IL-6, IL-12p70, and IL-10 in the supernatants and sera were determined using the appropriate DuoSet ELISA Development Systems (R&D Systems, Minneapolis, MN, USA) according to the manufacturer’s protocol.

### 2.8. Luciferase Assay

RAW264.7 cells were transfected with the pGL4.32 containing five copies of an NF-κB response element or pGL4.44 containing six copies of an AP-1 responsive element and the firefly luciferase reporter gene (luc2P) (Promega, Madison, WI, USA). The transfected RAW264.7 cells were plated into 96 well plates and incubated at 37 ℃ overnight. After changing the media, cells were pretreated with UWG, EWG, or DMBQ for 2 h and then stimulated with LPS for 6 h. Luciferase activity was measured using the Dual-Glo® luciferase assay system (Promega, Madison, WI, USA).

### 2.9. In vivo Experiments

Mice were randomly divided into normal, control, 0.3 g/kg UWG, 3 g/kg UWG, 0.3 g/kg EWG, and 3 g/kg EWG groups (*n* = 5 for the normal group and *n* = 13 for the control and treatment groups). UWG and EWG suspended in water were orally given to mice once daily for 4 weeks. Mice in the normal and control groups were given an equal amount of water. The control, UWG, and EWG groups were intraperitoneally injected with 1.3 mg/kg of LPS at the end of the experiment. After 1 h, the mice were anesthetized with ether, and blood was collected via cardiac puncture. Serum was obtained and stored at −20 ℃ for further analysis.

### 2.10. Real-Time RT PCR

RNA from the liver was isolated using Tri-RNA Reagent (Favorgen, Kaohsiung, Taiwan) and reverse-transcribed into cDNA using the High Capacity cDNA Reverse Transcription kit (Applied Biosystems, Foster, CA, USA). Real-time PCR was performed using SYBR Green Mix (Applied Biosystems, Foster, CA, USA) on a StepPlus One real-time PCR system (Applied Biosystems, Foster, CA, USA). Quantification of gene expression was determined by the standard curve calculation method. Target gene was normalized to GAPDH. 

### 2.11. Statistical Analysis

The data are presented as the mean ± SD. The two-tailed Student’s *t*-test or one-way analysis of variance was applied to compare the differences between groups. If the statistical analysis showed that the differences between multiple groups were significant, Tukey’s post-hoc test was used for further comparison. All statistical analyses were performed using IBM SPSS software, version 22.0 (Chicago, IL, USA). *P*-values less than 0.05 were considered statistically significant. 

## 3. Results

### 3.1. Analysis of DMBQ

The levels of DMBQ in UWG and EWG were measured. DMBQ was not detectable in UWG. EWG contained 0.244 ± 0.001 mg/g of DMBQ. 

### 3.2. Effects of UWG and EWG on Viability of Mouse Peritoneal Macrophages

Because EWG contains cytotoxic benzene metabolites, we first determined the cytotoxic response to UWG and EWG using mouse peritoneal macrophages. The conversion of tetrazolium into formazan via the mitochondrial succinate dehydrogenase system was used as a cell viability indicator [22]. Based on a report that the peak plasma concentration after oral consumption of fermented wheat germ extract ranges from 0.5 to 1 mg/mL [23], the highest concentration was set to 800 µg/mL of the extracts. Cell viability in UWG-treated cells increased compared to untreated cells (Figure 1A). Given the fact that primary cells do not divide ex vivo, these findings can be interpreted as the increased enzymatic cellular activity by UWG [24]. On the other hand, EWG did not show any increase in formazan production and increasing concentrations tended to decrease MTT reduction (Figure 1B). Additionally, the cytotoxic response of cells to DMBQ at concentrations corresponding to those in EWG (0–216 ng/mL) was measured. The highest concentration of DMBQ, which is equal to its concentration in 800 µg/mL EWG showed a 50% reduction in cell viability compared to the control (Figure 1C). This indicates that the lower enzymatic activity seen at higher concentrations of EWG may be attributable to DMBQ.

### 3.3. Effects of UWG and EWG on Inflammatory Enzyme Expression

We used an in vitro LPS-stimulated macrophage model to determine whether enzyme treatment affects the anti-inflammatory activity of UWG. First, we evaluated iNOS and COX2 protein expression using Western blotting. Both UWG and EWG decreased iNOS protein synthesis, with EWG being more potent (Figure 2). DMBQ at 0.049 µg/mL, which corresponds to the concentration found in 200 µg/mL of EWG, abrogated iNOS expression. COX2 protein expression was not affected by either extract treatment or by DMBQ. 

### 3.4. Effects of UWG and EWG on Pro-Inflammatory Cytokine Secretion 

We next examined the levels of TNF-α, IL-6, and IL-12 produced by LPS-stimulated macrophages. No inhibitory activity toward the secretion of TNF-α and IL-6 was found for UWG, EWG, or DMBQ (Figure 3A, B). However, UWG and EWG caused a dose-dependent inhibition of IL-12 secretion with equal potency, and DMBQ also inhibited IL-12 secretion (Figure 3C). These results indicate that UWG contains substances that specifically inhibit IL-12 production and that enzyme treatment does not affect this activity.

### 3.5. Effects of UWG and EWG on Anti-Inflammatory Protein Expression

Next, IL-10 secretion in LPS-stimulated macrophages was examined. UWG increased IL-10 production, as was previously shown, and EWG also increased IL-10 secretion but with less potency than UWG [20] (Figure 4A). DMBQ tended to decrease IL-10 secretion, indicating that it is likely responsible for the decreased activity of EWG to induce IL-10. We next examined the time course of HO-1 protein induction after EWG treatment in mouse peritoneal macrophages. HO-1 expression reached a maximum at 24 h (Figure 4B). Based on this time kinetics, we incubated macrophages with UWG, EWG, and DMBQ for 24 h. UWG stimulated HO-1 protein synthesis, and enzyme treatment increased this activity, but DMBQ did not induce HO-1 expression (Figure 4C). We further confirmed that UWG and EWG showed a similar pattern of HO-1 induction in the presence of LPS (Figure 4D). 

### 3.6. Effects of UWG and EWG on LPS-Induced Signaling Molecules

The NF-κB and MAPK-dependent AP-1 pathways are the earliest molecular channels that lead to the production of pro-inflammatory proteins in response to LPS [25]. Therefore, we investigated the effects of UWG, EWG, and DMBQ on IκBα degradation, which is a pre-requisite for NF-κB activation, and MAPK (p38, JNK, and ERK) phosphorylation, which is the upstream event of AP-1 activation. EWG potently inhibited IκBα degradation and JNK activation, and to a lesser degree, p38 and ERK activation (Figure 5A). None of these signaling molecules were affected by UWG and DMBQ. Next, we examined whether the transcriptional activity of NF-κB and AP-1 might be altered using a luciferase reporter gene assay. At a dose of 200 µg/mL, UWG and EWG significantly decreased NF-κB and AP-1 activity with EWG being more potent (Figure 5B,C). DMBQ did not display any effect on the transcriptional activity of these proteins.

### 3.7. Effects of UWG and EWG on the Nuclear Translocation of Nrf2 

After confirming the ability of UWG and EWG to induce HO-1 expression, we further explored EWG-induced nuclear import of nuclear factor erythroid 2-related factor (Nrf2), which occurs prior to HO-1 induction [26]. The nuclear migration of Nrf2 was measured at various time points. At 60 min after EWG treatment, a noticeable band of nuclear Nrf2 protein appeared in the Western blot of mouse peritoneal macrophages (Figure 6A). Based on this time kinetics, we compared the amount of nuclear Nrf2 in cells treated with UWG and EWG at 60 and 90 min. In keeping with the HO-1 response, EWG was more potent in stimulating the nuclear translocation of Nrf2 than UWG (Figure 6B).

### 3.8. Effects of UWG and EWG on Systemic Response Following Intraperitoneal Injection of LPS 

We also evaluated whether supplementation with UWG or EWG affected the systemic response to acute LPS challenge. The maximum oral dose of 3 g/kg was determined from published papers on fermented wheat germ extract [4,27]. Mice were fed UWG and EWG at doses of 0.3 or 3 g/kg for four weeks and then intraperitoneally stimulated with a sublethal dose of LPS. A decrease in serum TNF-α was observed in both the UWG and EWG groups (Figure 7A). EWG was effective even at the low dose, but there was no dose-dependent effect. Neither treatment showed any effect on serum IL-6 levels (Figure 7B). UWG was more effective than EWG in inducing IL-10 (Figure 7C), consistent with the in vitro findings. However, in the range of doses used in vivo, the lower dose of UWG showed a higher level of serum IL-10. These data indicate that compounds that interfere with IL-10 induction may compete with those that induce IL-10 at increasing doses of UWG. We also examined hepatic HO-1 gene expression, which rapidly responds to intraperitoneal LPS administration. Upregulation of HO-1 gene expression was observed in the 0.3 g/kg EWG group only (Figure 7D). As seen with IL-10 induction in the UWG group, compounds that induce HO-1 expression may be counteracted by other compounds present in the high-dose EWG group. 

## 4. Discussion

Wheat germ contains water-soluble anti-inflammatory ingredients and enzyme treatment facilitates the release of some of these ingredients. Our previous and current studies have shown that direct exposure of LPS-stimulated macrophages to UWG resulted in decreased IL-12 production and increased IL-10 production. These properties were not altered by enzyme treatment. Instead, enzyme treatment enhanced the downregulation of LPS-induced iNOS by UWG. HO-1 expression was also enhanced by enzyme treatment. Some of the changes brought by enzyme treatment could be attributed to DMBQ, but others seem to be related to the presence of unidentified byproducts in EWG.

The current study found that UWG was able to inhibit LPS-induced iNOS protein expression, which was not seen in the previous study. This difference appears to be related to the incubation temperature at which UWG was prepared. Previously, UWG was prepared at 60 ℃, and in this study, it was prepared at 30 ℃—the temperature at which the wheat germ’s endogenous enzymes or its microbial enzymes would likely act to generate the compounds that inhibit iNOS synthesis. More of these as-yet unknown compounds might be present in EWG. DMBQ is a potent inhibitor of iNOS expression but given the finding that UWG possesses this activity, DMBQ may not be entirely responsible for the increased downregulation of iNOS by EWG.

In addition to altering the effects on iNOS and IL-10 expression, enzyme treatment of wheat germ led to a greater inhibition of LPS-induced NF-κB and AP-1 activity and induction of Nrf2 activation and HO-1 expression in vitro. DMBQ did not show any effect on NF-κB, AP-1 and HO-1. Other ingredients generated by enzyme treatment must be responsible for these activities. Evidence shows that some flavonoids inhibit iNOS protein expression partly through induction of HO-1 [28]. Since wheat germ is a rich source of flavonoids, most of which exist in the form of glycosides, enzyme treatment may facilitate the release of flavonoids with similar activity.

The in vitro LPS-stimulated macrophage model is the most commonly used tool for screening natural products or drug candidates with anti-inflammatory activity. Neither UWG nor EWG affected the secretion of TNF-α in this model. However, LPS-induced serum elevation of TNF-α levels decreased in mice fed UWG or EWG. Such discrepancies can occur when natural products containing polysaccharides or glycosides are tested for their anti-inflammatory potentials. In vitro treatment of macrophages with these plant-derived polysaccharides or glycosides can stimulate the release of TNF-α through pattern-recognition receptor-mediated signaling [29,30,31]. Because of this, in vitro approaches sometimes fail to detect the ability of active polysaccharides to inhibit LPS-induced TNF-α secretion. However, when these polysaccharides and glycosides are orally administered, they can be metabolized by digestive enzymes or intestinal bacteria and induce biological responses. The inhibitory effect of EWG was observed at a lower dose compared with UWG, suggesting that the concentration of substances responsible for suppressing the secretion of TNF-α is increased by enzyme treatment. 

Since Hidvegi et al. standardized the amount of MBQ and DMBQ to the fermentation process for wheat germ, many studies have focused on the anti-cancer potential of the water extract from fermented wheat germ. In addition, Hidvegi et al. demonstrated the immunostimulatory activity of this extract, indicating that these opposing effects cannot be accounted for solely by the presence of quinones [4,27]. Recently, Zhang et al. raised the possibility that some of the protein fractions generated by fermenting wheat germ with yeast and lactic acid bacteria might contribute to the anti-cancer activity of the water extract of fermented wheat germ [6]. Therefore, whether quinones are the useful standard compounds for the development of wheat germ-derived functional foods needs to be reconsidered.

Fermentation has been the most common biotechnology to process wheat germ. However, chemical reactions involving microbial enzymes could further complicate the identification of active components. On the other hand, enzyme treatment could make the targets more selective, which could help to identify the active compounds. Our study suggests that when it comes to the development of anti-inflammatory wheat germ extract, DMBQ is the least likely candidate to be the quality marker for manufacture. A limitation of our study is that we did not characterize changes in chemical constituents following enzyme treatment except for DMBQ. Because wheat germ is rich in as-yet unidentified compounds, it will be a challenging task to complete such an analysis. Future study is required to shed some light on the active components of anti-inflammatory water-soluble wheat germ extract. 

## 5. Conclusions

In conclusion, we used enzyme treatment to generate DMBQ-enriched wheat germ and compared the anti-inflammatory activities of the water extracts from untreated and enzyme-treated wheat germ in vitro and in vivo. UWG contains anti-inflammatory ingredients; enzyme treatment facilitates the release of more anti-inflammatory compounds as DMBQ does not account for the entirety of EWG’s activity. Our results provide evidence that enzyme treatment is a useful biotechnology tool for extracting water-soluble components from wheat germ and may be helpful in the identification of active components in the future. 

## Figures and Tables

**Figure 1 nutrients-11-02490-f001:**
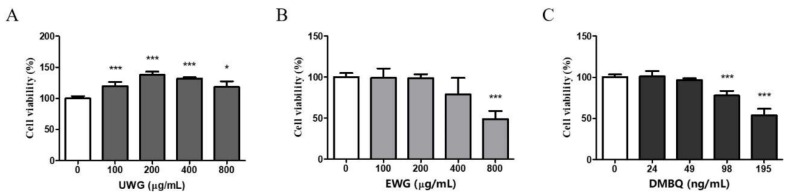
Effects of untreated wheat germ extract (UWG) and enzyme-treated wheat germ extract (EWG) on cell viability. Mouse peritoneal macrophages were cultured with UWG (**A**), EWG (**B**), or 2,6-dimethoxy-benzoquinone (DMBQ) (**C**) for 24 h, and cell viability was determined using the MTT assay. Bars represent the percentage of control cells (0 µg/mL). * *p* <0.05, *** *p* <0.005 vs. control cells, (*n* = 4).

**Figure 2 nutrients-11-02490-f002:**
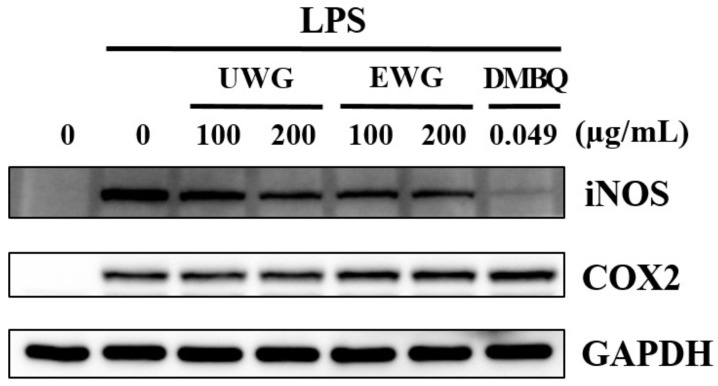
Effects of UWG and EWG on the protein expression of inducible nitric oxide synthase (iNOS) and cyclooxygenase (COX)2. Mouse peritoneal macrophages were exposed to UWG, EWG, or DMBQ and simultaneously with 100 ng/mL lipopolysaccharide (LPS) for 24 h. The protein expression of iNOS and COX2 was analyzed using Western blotting. GAPDH was used as an internal control. One of three independent experiments is shown.

**Figure 3 nutrients-11-02490-f003:**
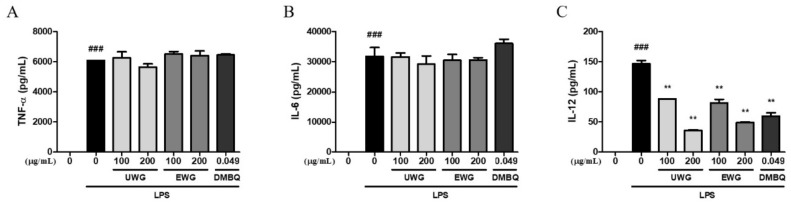
Effects of UWG and EWG on tumor necrosis factor (TNF)-α, interleukin (IL)-6, and IL-12. Mouse peritoneal macrophages were exposed to UWG, EWG, or DMBQ and simultaneously with LPS for 24 h. The levels of TNF-α (**A**), IL-6 (**B**), and IL-12 (**C**) in the supernatant were analyzed using ELISA. ### *p* <0.005 vs. LPS (-), ** *p* <0.01 vs. LPS (+), (*n* = 3).

**Figure 4 nutrients-11-02490-f004:**
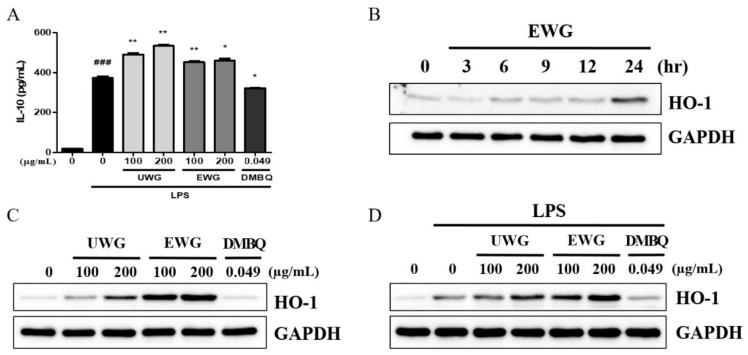
Effects of UWG and EWG on the anti-inflammatory proteins IL-10 and heme oxygenase (HO)-1 in mouse peritoneal macrophages. (**A**) Cells were stimulated with LPS in the presence of UWG, EWG, or DMBQ for 24 h, and the levels of IL-10 in the supernatant were analyzed using ELISA. ### *p* <0.005 vs. LPS (-), * *p* <0.05, ** *p* <0.01 vs. LPS (+), (*n* = 3). (**B**–**D**) Analysis of HO-1 synthesis via Western blotting. (**B**) Cells were cultured with EWG for the indicated time period. (**C**) Cells were cultured with UWG, EWG, or DMBQ for 24 h. (**D**) Cells were exposed to UWG, EWG, or DMBQ and simultaneously stimulated with LPS. One of three independent experiments is shown.

**Figure 5 nutrients-11-02490-f005:**
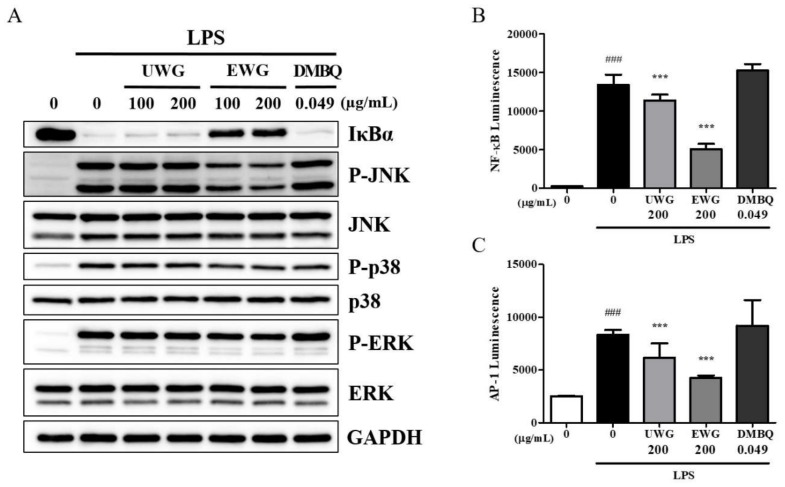
Effects of UWG and EWG on LPS-induced NF-κB, MAPK, and AP-1 activation. (**A**) Mouse peritoneal macrophages were pretreated with UWG, EWG, or DMBQ for 1 h and then stimulated with LPS for 15 min. Whole cell lysates were prepared and subjected to Western blotting. One of three independent experiments is shown. (**B**–**C**) RAW264.7 cells were transfected with an NF-κB- or AP-1-dependent reporter gene. Cells were pretreated with UWG, EWG, or DMBQ for 2 h and then stimulated with LPS for 6 h. Luciferase activity was measured using the Dual- Glo® luciferase assay system. ### *p* <0.005 vs. LPS (-), *** *p* <0.005 vs. LPS (+), (*n* = 3).

**Figure 6 nutrients-11-02490-f006:**
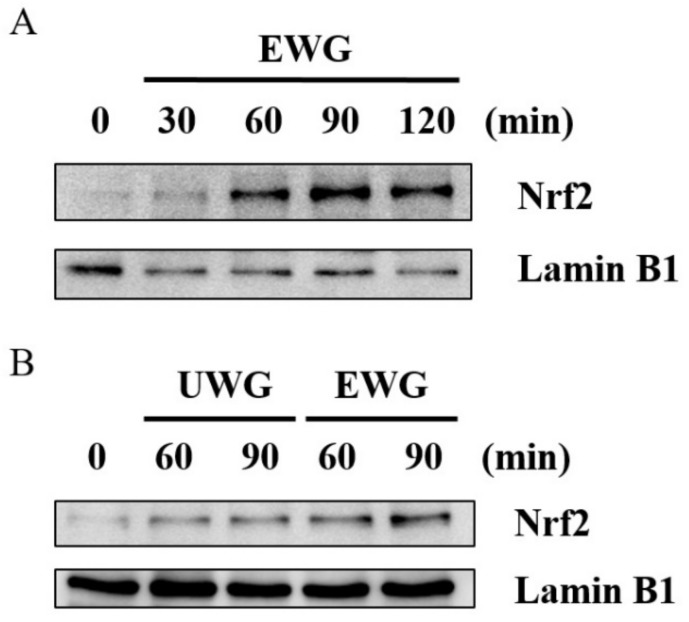
Effects of UWG and EWG on Nrf2 activation. (**A**) Mouse peritoneal macrophages were cultured with EWG for the indicated time periods. (**B**) Mouse peritoneal macrophages were cultured with 200 µg/mL of UWG or EWG for 60 or 90 min. For all experiments, nuclear proteins were prepared and the levels of Nrf2 were analyzed using Western blotting with lamin B1 as an internal control. One of three independent experiments is shown.

**Figure 7 nutrients-11-02490-f007:**
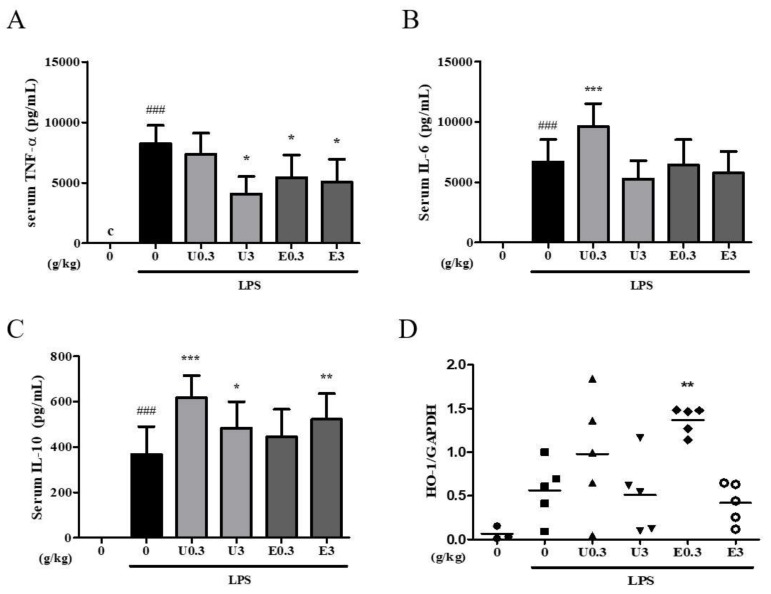
Effects of UWG and EWG on the systemic response to LPS. Following treatment with UWG (U) or EWG (E) for 4 weeks, mice were intraperitoneally injected with LPS (1.3 mg/kg), and serum and the liver were obtained one hour later. (**A–C**) The cytokine levels were measured using ELISA (*n* = 5 for the normal group; *n* = 13 for the control and treatment groups). (**D**) Liver HO-1 gene expression levels were determined using real-time RT-PCR (*n* = 3–5). ### *p* <0.005 vs. LPS (-), * *p* <0.05, ** *p* <0.01, *** *p* <0.005 vs. LPS (+)

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
