# Peer review of "Enzyme Treatment Alters the Anti-Inflammatory Activity of the Water Extract of Wheat Germ In Vitro and In Vivo"

_nutrients, 2019, doi:10.3390/nu11102490_

Round 1
Reviewer 1 Report
This is an interesting study and a rather well written MS.
Some points to improve the MS are:
the authors are asked to re-write the Introduction around wheat germ and not inflammation! The topic of their MS is wheat germ so some information on the origins and the chemical composition (as found in previous studies) of wheat germ must be provided in the introduction. The authors are kindly asked also to include in their introduction, other studies that have shown anti-inflammatory properties of wheat related products and by-products. Suggested references: https://www.mdpi.com/2304-8158/8/5/171 and https://www.sciencedirect.com/science/article/pii/S2212429218305388 in Fig. 1, in the case of UWG, why viability in the range of 130% was observed? What is the underlying reason for this? A better justification needs to be provided especially for the case of UWG 200μg/ml. the first paragraph of discussion is out of place (lines 319-330); this paragraph should be moved after the authors' discussion of their own results. Since this paper has been submitted to Molecules, it would be worth to have a full chemical analysis of UWG and EWG, at the beginning of the results section (paragraph 3.1).I would suggest major revision, happy to review the revised MS.
Author Response
We thank reviewers for the comments and suggestions. We have highlighted the specific changes in the revised manuscript and provide our response in this cover letter (see below). We hope that changes made in our revised manuscript have addressed the concerns raised.
1) The authors are asked to re-write the Introduction around wheat germ and not inflammation! The topic of their MS is wheat germ so some information on the origins and the chemical composition (as found in previous studies) of wheat germ must be provided in the introduction.
Response) As suggested by the reviewer, we reorganized the introduction. We focused on wheat germ and have shortened the paragraph on inflammation. The origin of wheat germ was specified in the Materials and Method (please see line 95). We added more information on the chemical composition of wheat germ (please see the first paragraph of the Introduction section). of wheat. Thank you for improving our manuscript.
2) The authors are kindly asked also to include in their introduction, other studies that have shown anti-inflammatory properties of wheat related products and by-products. Suggested references: https://www.mdpi.com/2304-8158/8/5/171 and https://www.sciencedirect.com/science/article/pii/S2212429218305388
Response) As suggested by the reviewer, we searched for studies that have shown anti-inflammatory properties of wheat or other cereal related products and found a few papers including the reviewer’s suggestions. We included these studies in the revised manuscript (please see lines 72-73).3) In Fig. 1, in the case of UWG, why viability in the range of 130% was observed? What is the underlying reason for this? A better justification needs to be provided especially for the case of UWG 200μg/ml.
Response) Thank you for your kind comment. We did not pay a close attention to the significance of these data before you pointed it out. Because we used mouse peritoneal macrophages, not a cell line, these cells do not divide ex vivo. Therefore, the increased MTT value by untreated wheat germ extract (UWG) must represent the increased enzymatic activity that reduces tetrazolium to formazan. This indicates that UWG stimulates cell’s metabolism in vitro while enzyme treated wheat germ extract (EWG) does not. We mentioned these findings and interpretation in the 3.2 section (please see lines 197-224).4) The first paragraph of discussion is out of place (lines 319-330); this paragraph should be moved after the authors' discussion of their own results.
Response) As suggested, we relocated the first paragraph of Discussion to the end of Discussion (please see lines 346-354). Thank you for improving the quality of our manuscript.5) Since this paper has been submitted to Molecules, it would be worth to have a full chemical analysis of UWG and EWG, at the beginning of the results section (paragraph 3.1).
Response) We inform you that the journal we have submitted to is Nutrients. However, the reviewer makes a valid point regardless of the journal type. We certainly understand that it is worth to have a full chemical analysis of UWG and EWG. As with other natural products, wheat germ is rich in as-yet unknown components. As stated in the Introduction, DMBQ and MBQ are the quality markers for the manufacture of wheat germ extract. The aim of the current study is to clarify that the anti-inflammatory activity of water-soluble extract of processed or unprocessed wheat germ does not depend on quinones. We thank to the reviewer for the relevant suggestion and we will consider it for further investigation. We have addressed this point at the end of the discussion. We are planning to carry out bioactivity-guided fraction study and search for active components that represent the anti-inflammatory activity of wheat germ.
Reviewer 2 Report
The authors of ‘Enzyme treatment alters the anti-inflammatory activity of the water extract of wheat germ in vitro and in vivo’ investigated whether Celluclast 1.5L treatment altered the anti-inflammatory activity of wheat germ.
All experiments on this article are designed properly, and each data is statistically analyzed by a suitable method. Several results are highly interesting, which showed that EWG might be more effective to suppress the cellular inflammatory response.
However, most results on this article implicated that DMBQ was not critical for the activity of EWG, and the authors did not investigate what components were responsible for its activity at all. I would recommend their re-submission if the authors could identify the critical components which truly contribute to the activity of EWG.
Author Response
The authors of ‘Enzyme treatment alters the anti-inflammatory activity of the water extract of wheat germ in vitro and in vivo’ investigated whether Celluclast 1.5L treatment altered the anti-inflammatory activity of wheat germ. All experiments on this article are designed properly, and each data is statistically analyzed by a suitable method. Several results are highly interesting, which showed that EWG might be more effective to suppress the cellular inflammatory response. However, most results on this article implicated that DMBQ was not critical for the activity of EWG, and the authors did not investigate what components were responsible for its activity at all. I would recommend their re-submission if the authors could identify the critical components which truly contribute to the activity of EWG.
Response) The reviewer makes a valid point. One of the limitations of our study is that we did not identify the critical components that were responsible for the anti-inflammatory activity of wheat germ extract. We have addressed this issue at the end of the discussion. As with other natural products, wheat germ is rich in as-yet unknown components. As stated in the Introduction, DMBQ and MBQ are currently the standard quality markers for the manufacture of wheat germ extract. The aim of the current study is to prove that the anti-inflammatory activity of water-soluble extract of processed or unprocessed wheat germ does not depend on quinones. We will consider the reviewer’s suggestion for further investigation. We are planning to carry out bioactivity-guided fraction study and search for active components that represent the anti-inflammatory activity of wheat germ and be used for the quality control of wheat germ processing.
Round 2
Reviewer 1 Report
The MS is fit now for publication.
Reviewer 2 Report
I would recommend their submission to other journals, or re-submission if the authors could identify the critical components which truly contribute to the activity of EWG.